# Prognostic Factors for Resolution Delay of Lower Urinary Tract Symptoms in Patients with Prostate Cancer after Low-Dose-Rate Brachytherapy

**DOI:** 10.3390/cancers15164048

**Published:** 2023-08-10

**Authors:** Tomoki Taniguchi, Makoto Kawase, Keita Nakane, Masahiro Nakano, Koji Iinuma, Daiki Kato, Manabu Takai, Yuki Tobisawa, Takayuki Mori, Hirota Takano, Tomoyasu Kumano, Masayuki Matsuo, Takayasu Ito, Takuya Koie

**Affiliations:** 1Department of Urology, Ogaki Municipal Hospital, 4-86 Minaminokawacho, Ogaki, Gifu 5038502, Japan; tomokidbx@gmail.com; 2Department of Urology, Graduate School of Medicine, Gifu University, 1-1 Yanagido, Gifu 5011194, Japan; kawase.makoto.g5@f.gifu-u.ac.jp (M.K.); nakane.keita.k2@f.gifu-u.ac.jp (K.N.); iinuma.koji.s0@f.gifu-u.ac.jp (K.I.); kato.daiki.m2@f.gifu-u.ac.jp (D.K.); takai.manabu.a5@f.gifu-u.ac.jp (M.T.); tobisawa.yuki.a7@f.gifu-u.ac.jp (Y.T.); 3Department of Urology, Gifu Prefectural General Medical Center, 4-6-1 Noisiki, Gifu 5008717, Japan; nakano-m@juno.ocn.ne.jp; 4Department of Radiology, Graduate School of Medicine, Gifu University, 1-1 Yanagido, Gifu 5011194, Japan; mori.takayuki.b4@f.gifu-u.ac.jp (T.M.); takano.hirota.w2@f.gifu-u.ac.jp (H.T.); kumano.tomoyasu.k0@f.gifu-u.ac.jp (T.K.); matsuo.masayuki.e0@f.gifu-u.ac.jp (M.M.); 5Center for Clinical Training and Career Development, Graduate School of Medicine, Gifu University, 1-1 Yanagido, Gifu 5011194, Japan; ito.takayasu.v9@f.gifu-u.ac.jp

**Keywords:** prostate cancer, brachytherapy, lower urinary tract symptoms, overactive bladder symptom score

## Abstract

**Simple Summary:**

Chronological changes in the overactive bladder symptom score (OABSS) and the time-to-resolution of the OABSS were evaluated in 237 patients with prostate cancer who underwent low-dose-rate brachytherapy (LDR-BT) with iodine-125. The patients were divided into two groups: the OABSS resolution group and the resolution delay group. The OABSS in both groups worsened at 3 months following operation and gradually recovered at 9 months; however, the OABSS in the resolution delay group tended to worsen again after that. In the multivariate analysis, preoperative OABSS and the change from baseline to maximal OABSS were associated with OABSS resolution. To the best of our knowledge, this is the first study to evaluate the delayed resolution of storage symptoms.

**Abstract:**

Urinary storage symptoms after low-dose-rate brachytherapy (LDR-BT) with iodine-125 have been noted to be less likely to improve to baseline compared to voiding symptoms. This study aimed to evaluate the chronological changes in the overactive bladder symptom score (OABSS) and the time-to-resolution of OABSS in patients undergoing LDR-BT. Patients with prostate cancer who underwent LDR-BT at Gifu University Hospital were enrolled. The OABSS was evaluated before and after LDR-BT. Patients were divided into the OABSS resolution and resolution delay groups, and the association between OABSS resolution delay and clinicopathological covariates was evaluated. In total, 237 patients were enrolled in this study, with a median follow-up of 88.3 months. The OABSS in both groups worsened at 3 months following operation and gradually recovered at 9 months; however, the OABSS in the resolution delay group tended to worsen again after that. In the multivariate analysis, preoperative OABSS and the change from baseline to maximal OABSS were associated with OABSS resolution. To our knowledge, this is the first study to evaluate the delayed resolution of OABSS after LDR-BT in patients with prostate cancer. A low baseline OABSS and significant changes in the OABSS from baseline were independent predictors of delayed OABSS resolution.

## 1. Introduction

Prostate cancer (PCa) has the highest incidence rate among men in Japan [1]. According to the database of the National Cancer Center in Japan, approximately 94,000 patients were newly diagnosed with PCa in 2019, and 12,000 men deceased from PCa in 2020 [2]. Most patients with localized PCa are treated with radical prostatectomy (RP), external beam radiation therapy (EBRT), or brachytherapy as definitive therapy [3]. Among them, low-dose-rate brachytherapy (LDR-BT) with iodine-125 has been considered a curative therapy for low-risk PCa [4]. Patients with intermediate- or high-risk PCa are treated with LDR-BT combined with EBRT and/or androgen deprivation therapy (ADT) [4]. Regardless of the type of therapy, patients with localized PCa generally have a favorable prognosis, with a slow but not rapidly progressive course in most cases [5,6]. Therefore, late complications after the treatment of localized PCa, as well as oncologic outcomes, are considered very important factors in the choice of treatment modality when considering the patient’s quality of life. Regarding radiotherapy, the need to increase the radiation dose to improve oncological outcomes and the possibility of increased acute and/or long-term complications, such as genitourinary (GU) and gastrointestinal toxicities, should be considered [4,7]. The rates of Grade 2 and Grade ≥3 late GU toxicity by the Radiation Therapy Oncology Group (RTOG) associated with LDR-BT were 21% and 2–3%, respectively [4]. 

Regarding urination in patients with PCa undergoing LDR-BT, the total International Prostate Symptom Score (IPSS) and quality of life due to urinary symptoms (IPSS-QOL) scores increased compared to baseline at 3 months following operation and returned to baseline by 36 months, whereas the overactive bladder symptom score (OABSS) did not return to baseline even at 36 months after LDR-BT [8]. In our previous study, we reported that the IPSS, IPSS-QOL, and OABSS improved over time and returned to baseline values 18–36 months after LDR-BT [9]. We also found that IPSS and IPSS-QOL remained virtually unchanged compared to baseline values after a 5-year follow-up period, whereas OABSS worsened significantly [9]. These results suggest that LDR-BT is more likely to affect urinary storage than voiding symptoms, which may persist for longer periods.

Here, we aimed to investigate the predictive factors influencing the delayed resolution of OABSS in patients with PCa who underwent LDR-BT.

## 2. Materials and Methods

### 2.1. Patient Selection

We conducted a retrospective cohort study of consecutive patients with clinical T1c/T2/T3a PCa, according to the 7th American Joint Committee on Cancer Staging Manual [10], who underwent LDR-BT at Gifu University Hospital between August 2000 and December 2015. The enrolled patients were classified into very low-, low-, favorable intermediate-, poor intermediate-, high-, and very-high-risk groups according to the classification model proposed by the National Comprehensive Cancer Network (NCCN) guidelines (version 4, 2018) [11]. 

An institutional review board of Gifu University authorized the present study (approval number: 2020-210). Since this is a retrospective study, informed consent was waived. According to the regulations of the Japanese Ethics Committee and ethical guidelines, retrospective and observational studies using existing data and other materials do not require written consent because the results of these studies have already been released to the public. Details of this study, which are available only in Japanese, can be found at https://www.med.gifu-u.ac.jp/visitors/disclosure/docs/2020-210.pdf (accessed on 6 June 2023).

### 2.2. Treatment

The enrolled patients received either a linked seed using the ProLink^®^ delivery system (C. R. Bard, Inc., Murray Hill, NJ, USA) or loose ^125^I radioactive seeds (Oncoseed, Nihon Mediphysics, Tokyo, Japan), which were implanted by real-time transrectal ultrasound (TRUS)-guided transperineal technique [12]. The prescribed minimum peripheral doses were 145 Gy for patients who received LDR-BT alone and 104 Gy for those who received a combination of LDR-BT and EBRT. EBRT was performed on the prostate and seminal vesicles within 1 month of LDR-BT (40 Gy in 2 Gy fractions). Patients with very low- or low-risk PCa who had a pre-treatment prostate volume (PV) of >50 mL received neoadjuvant ADT for at least 3 months to decrease PV. Patients with favorable and unfavorable intermediate-risk PCa were treated with ADT for 9 months in combination with LDR-BT and/or EBRT. Patients with high- or very-high-risk PCa underwent 24 months of ADT before and after combined LDR-BT and EBRT. All enrolled patients were administered an α1-blocker after LDR-BT to reduce symptoms, including urinary retention and lower urinary tract symptoms (LUTS). All patients underwent preplanning using a modified peripheral loading technique prior to seed implantation [13].

### 2.3. Post-Dosimetric Evaluation

The latest American Association of Physicists in Medicine Task Group 43 formalism and Variseed version 7.1 (Varian Medical Systems, Palo Alto, CA, USA) were used for treatment planning and post-implant dosimetry. One month after LDR-BT, post-implant dosimetry was performed using computed tomography (CT) and magnetic resonance imaging (MRI). CT was performed using a CT scanner (LightSpeed Ultra 16/Discovery CT 750HD; GE Healthcare, Milwaukee, WI, USA) with a 16 or 64 detector array. MRI was performed using a 5-channel SENSE cardiac coil with easy breathing, a slice thickness of 3 mm, and no cross-gap (Intera Achieva 1.5 T/Intera Achieva Nova Dual 1.5 T Pulsar; Philips Medical Systems, Eindhoven, The Netherlands). The following dose parameters were used and analyzed in this study: the minimum dose received by 90% of the prostate gland (D90), percentage of the target volume receiving a minimum of 100% of the prescribed dose (V100), minimum dose received by 30% of the urethral volume (UD30), rectal volume receiving 100% of the prescribed dose (RV100), and rectal volume receiving 150% of the prescribed dose (RV150).

### 2.4. Follow-Up Schedule

IPSS, IPSS-QOL, OABSS, uroflowmetry (UFM), and post-voiding residual volume (PVR) were evaluated before LDR-BT; 1, 3, 6, 9, 12, 18, and 24 months after LDR-BT; and annually thereafter.

### 2.5. Endpoints and Statistical Analysis

The primary endpoints were the chronological changes and time-to-resolution of the OABSS after LDR-BT. The secondary endpoint was to analyze the relationship between delayed resolution of OABSS and clinicopathological covariates. Patients were divided into OABSS resolution and resolution delay groups. Resolution of the OABSS was defined as the return of the OABSS to baseline scores after LDR-BT. Chronological changes in OABSS were represented by line graphs with median and interquartile range (IQR) values. Data analysis was performed using JMP Pro 16 software (SAS Institute Inc., Chicago, IL, USA). Continuous variables were analyzed using the Wilcoxon signed-rank test, and categorical variables were analyzed using Fisher’s exact test. The Kaplan–Meier method was used for estimating the resolution of OABSS and assessing the proportion of OABSS resolution. The resolution of OABSS in each group was analyzed using the log-rank test. Multivariate analysis was performed using the Cox proportional hazards model. All *p*-values were two-tailed, with a *p*-value < 0.05 considered statistically significant. 

## 3. Results

### 3.1. Patient Characteristics

Patients with PCa and lymph node metastases, distant metastases, prior transurethral resection of the prostate, maximal flow velocity (Qmax) <10 mL/s, or missing OABSS values up to 36 months after LDR-BT were excluded from the study. Finally, 237 patients met the inclusion criteria. Table 1 lists the patient characteristics and prescribed dose data related to LDR-BT for the OABSS and delayed resolution groups. The median age, preoperative OABSS, maximum change in OABSS from baseline, and follow-up period for all patients were 66 years, three, four, and 88.3 months, respectively. Compared to the OABSS resolution group, the delayed resolution group was statistically younger and had lower pre-treatment prostate-specific antigen (PSA) levels, a lower preoperative OABSS, and greater change from baseline to maximum OAB

### 3.2. Chronological Changes in OABSS

The chronological changes from baseline in the OABSS after LDR-BT are shown in Figure 1. For all patients, the OABSS increased temporarily at 3 months after LDR-BT and decreased gradually at 9, 48, and 96 months following operation. 

Figure 2 shows the time-series changes in OABSS after LDR-BT for the two groups: OABSS resolution and resolution delay groups. Both groups showed a temporary increase in OABSS 3 months after LDR-BT and a trend toward improvement at 9 months. However, the OABSS in the delayed resolution group remained elevated thereafter and never recovered to baseline. In contrast, in the OABSS resolution group, OABSS did not worsen after 9 months of LDR-BT.

### 3.3. Proportion of OABSS Resolution

Figure 3 shows the proportion of patients in whom the OABSS improved to baseline. Almost half of the patients showed an improved OABSS at baseline by 20 months after LDR-BT. Thereafter, the proportion of patients with OABSS resolution gradually increased, reaching approximately 80% at 100 months after LDR-BT. 

Figure 4 shows the chronological changes in the rate of OABSS resolution in the two groups based on the difference between the preoperative OABSS and the maximum OABSS from baseline. Patients were divided into two groups based on the median values of preoperative OABSS and the difference between the maximum OABSS and baseline. With respect to preoperative OABSS, the OABSS resolution rates at 1, 3, and 5 years were 38.8%, 60.2%, and 68.2%, respectively, in the group with preoperative OABSS <4 and 53.2%, 73.4%, and 79.6%, respectively, in the group with OABSS ≥4 (*p* = 0.018; Figure 4A). When divided by change in OABSS from baseline to maximum, the 1-, 3-, and 5-year OABSS resolution rates were 61.7%, 80.5%, and 87.0%, respectively, in the group with a change of <5 compared to 30.3%, 53.2%, and 61.1%, respectively, in the group with a change of ≥5 (*p* < 0.001; Figure 4B).

The results of the multivariate Cox proportional hazards regression analysis for OABSS resolution are shown in Table 2. The preoperative OABSS and the change from baseline to maximum OABSS had a significant impact on OABSS resolution.

## 4. Discussion

LDR-BT, EBRT, or RP is a standard primary curative radical treatment for localized PCa [4,14]. The oncological outcomes of LDR-BT for PCa are considered comparable to those of EBRT and RP [15]. In general, biochemical recurrence (BCR) after LDR-BT is defined as an increase in PSA levels from nadir to 2 ng/mL based on Phoenix’s definition and EBRT [16]. Regarding oncologic outcomes of LDR-BT for PCa in Japan, biological recurrence-free survival rates were reported to be 92–98%, 84–97%, and 70–95% in the low-, intermediate-, and high-risk groups, respectively [17]. Similarly, Lazarev et al. [14] found a 17-year cancer-specific survival rate of 97% in patients treated with LDR-BT, indicating an excellent antitumor effect of LDR-BT for localized PCa. On the other hand, approximately 90% of patients who underwent LDR-BT developed GU toxicity, which is problematic because it worsens the postoperative quality of life of patients with PCa [18]. According to the Nationwide Japanese Prostate Cancer Outcome Study of Permanent Iodine-125 Seed Implantation (J-POPS) conducted in Japan, the incidence of acute and late GU toxicity after LDR-BT was 35–67% and 22–55%, respectively, and another reported that the incidence of severe urinary toxicity increased throughout the follow-up period and did not reach a plateau [18,19]. In addition, in the J-POPS study, older age, higher PV, higher preoperative IPSS, and alcohol consumption were significant independent predictors of acute GU toxicity, whereas higher PV, lower V100, higher preoperative IPSS, and a history of rectal cancer were significant independent predictors of late GU toxicity in a multivariate analysis [17]. In the current study, PV, dosimetric parameters, and preoperative IPSS were not associated with late GU toxicity for storage symptoms. Therefore, GU toxicity, especially LUTS, after seed implantation is associated with bladder ischemia and inflammation [20]. These reports suggest that patients with PCa who undergo LDR-BT may experience severe urinary toxicity for an extended period after surgery. 

A relatively large number of previous studies examining chronological changes regarding LUTS after LDR-BT have been reported [8,9,18,19,21,22,23,24,25]. Onishi et al. [8] reported that patients who experienced acute dysuria 3 months after LDR-BT gradually improved over time and returned to baseline after 36 months. In a study conducted using IPSS on changes in LUTS after LDR-BT, defining an increase of ≥5 points from nadir as a urinary symptom flare, it was reported that late LUTS symptoms could worsen in approximately half of the patients within 5 years after LDR-BT [22]. IPSS resolution was achieved in 72.2–82.1% and 83.3–91.9% of patients with PCa at 12 and 24 months after LDR-BT, respectively [23,26]. When compared with the preoperative IPSS, IPSS disappearance at 24 months was 39.4%, 59.5%, and 83.3% in mild, moderate, and severe cases, respectively, and longer IPSS resolution times were required in patients with lower preoperative scores [23,26,27]. However, there are few studies on the resolution rate of urinary storage symptoms using the OABSS [23,24,28]. Sakayori et al. reported OABSS resolution rates of 71.2% and 81.0% at 12 and 36 months, respectively, with a median preoperative OABSS score of 3 and a median OABSS change point from baseline to the maximum of 2 [23]. Multivariate analysis suggested that a PV ≥25 mL may delay the resolution of OABSS [23]. In the present study, the OABSS resolution rate was 47.3% at 12 months and 67.9% at 36 months, which was relatively poor compared to the results of previous studies. At our institution, patients with severe LUTS such as Qmax <10 mL/s, high IPSS, or high OABSS were excluded from LDR-BT, suggesting that these results may be due to selection bias. Another study comparing LUTS after LDR-BT and EBRT reported that LDR-BT was associated with a significantly higher incidence of long-term urgency symptoms, even 3 years following operation, than EBRT, independently of age and physical function [28]. It has also been reported that patients with PCa with a low pre-LDR-BT OABSS required a longer time to resolve OABSS following operation and that patients with higher preoperative OABSS improved to baseline at a higher rate than those with lower preoperative OABSS [24,25]. In the present study, the OABSS of patients with delayed resolution gradually worsened 3 months after LDR-BT and did not improve thereafter, although the delayed resolution of OABSS may be influenced by age-related changes. In this study, OABSS worsened only in the delayed resolution group 6 years after treatment, and the reason for this was unclear; however, changes in bladder function after LDR-BT could be one cause, and the results of this study suggest that age is not associated with a worsening OABSS. Therefore, it is possible that radiation therapy caused some damage to the normal tissues of the bladder. However, it is difficult to determine when and to what extent bladder tissue was affected in patients who received LDR-BT. In fact, there were no reports that examined in detail the effects on normal bladder tissue after radiotherapy. Further studies may be necessary in the future. In addition, we found that a lower OABSS before LDR-BT and a greater change in OABSS from baseline to the maximum were significantly associated with a delayed resolution of OABSS. The results of our study suggest that the use of the OABSS may help identify eligible patients with PCa who are appropriate for LDR-BT. If the OABSS changed significantly from baseline after LDR-BT, it seemed necessary to inform the patient that conservative symptoms might continue for some time.

Previous reports have shown an association between GU toxicity after LDR-BT and the delayed resolution of OABSS, and combination therapy with LDR-BT and EBRT has been associated with late GU toxicity and LUTS [8,17,19,21,23,24]. Tanaka et al. reported that the incidence of urinary frequency and urgency of GU toxicity were significantly higher in patients treated with LDR-BT alone at 3 and 12 months following operation; however, there was no difference in the incidence of GU toxicity at 24 and 36 months [19]. Similarly, patients treated with a combination of LDR-BT and EBRT showed a worsening OABSS at 24 and 36 months following operation, but this was not seen in those treated with LDR-BT alone [8]. Uematsu et al. [21] also showed that the urinary frequency 3 months after LDR-BT, using a frequency volume chart, was higher in combination therapy with EBRT. This may be due to the higher doses to the urethra and bladder in patients who received combined LDR-BT and EBRT than in those who received LDR-BT alone. Therefore, BED and UD30 are known to affect LUTS after LDR-BT [17,21,23,24]. A large American cohort study reported that BED, combined EBRT and LDR-BT, ADT, patient age, and PV did not affect long-term LUTS [27]. In a multivariate analysis, the total dose did not affect long-term LUTS development, although the higher the BED, the slower the return to the baseline score 2 years following implantation [28]. It has also been reported that ADT use, preoperative IPSS, IPSS elevation from baseline, the presence of early postoperative LUTS, and V150 were independent predictors of the development of late GU toxicity [29]. In the present study, which allowed for long-term observation, OABSS resolution was not associated with additional EBRT treatment, concomitant ADT, BED, or UD30. Although these factors might temporarily worsen LUTS in the early postoperative period, OABSS tended to improve gradually over time, suggesting that they might not affect OABSS ≥ 4 years following operation.

This study had several limitations. First, potential bias may be inherent because it was a retrospective, single-center, non-randomized study. Secondly, the number of enrolled patients was relatively small; therefore, caution must be exercised when interpreting the findings of this study. Third, because the decision to resolve the OABSS was based on the judgment of the physician or individual patient, the interpretation of the results may not have been based on certain criteria, suggesting that the results may have differed from those obtained when the diagnosis was based on clear criteria. In addition, the IPSS and OABSS results were obtained from individual patients, suggesting that the assessment of LUTS may not be consistent [30]. Fourth, there is heterogeneity of the radiotherapy type in this cohort: linked seeds or loose seeds; LDR-BT monotherapy or combination therapy with EBRT. This leads to a potential bias and should be avoided in future prospective studies. Finally, we did not collect data on the use of anticholinergics or β3-adrenoceptor agonists after LDR-BT. Therefore, the effects of these drugs on the improvement of urinary symptoms were not considered in the present study. 

## 5. Conclusions

To the best of our knowledge, this is the longest follow-up study to evaluate chronological changes in OABSS in patients with PCa after LDR-BT. In addition, this study identified independent factors influencing OABSS resolution. A low OABSS before LDR-BT and a large change in OABSS from baseline were independent predictors of delayed OABSS resolution.

## Figures and Tables

**Figure 1 cancers-15-04048-f001:**
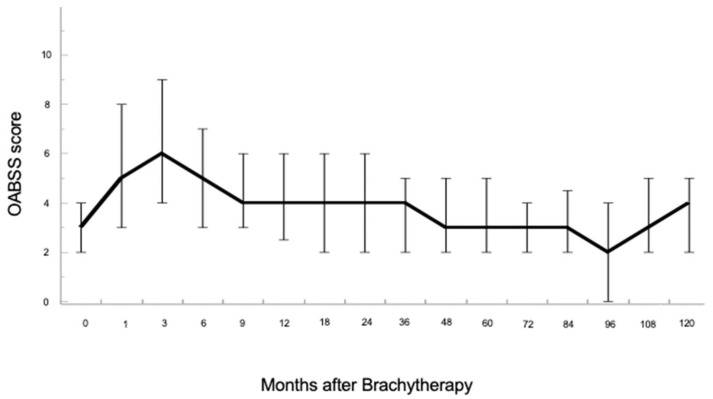
Chronological changes in overactive bladder symptom score (OABSS) were shown in a linear mixed-effects model. After low-dose-rate brachytherapy (LDR-BT) with iodine-125, OABSS scores temporarily increased at 3 months, followed by OABSS improvement to baseline values at 48 months.

**Figure 2 cancers-15-04048-f002:**
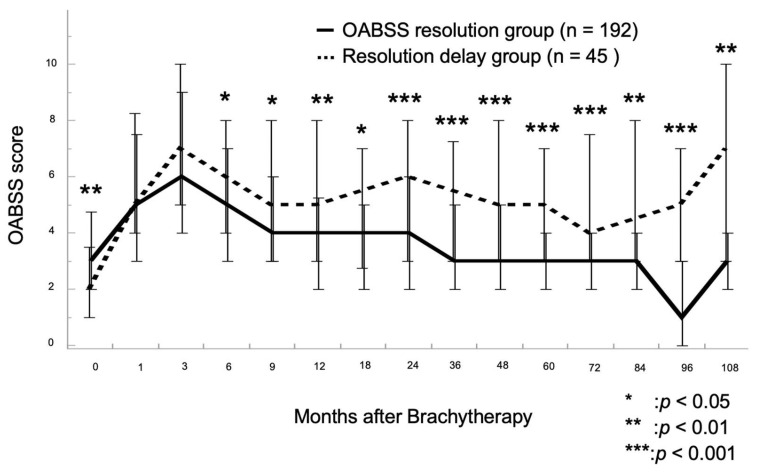
Chronological changes in the overactive bladder symptom score (OABSS) in the OABSS resolution and delayed resolution groups were evaluated with a linear mixed-effects model. In both groups, OABSS temporarily increased at 3 months after low-dose-rate brachytherapy (LDR-BT) with iodin-125 and decreased at 9 months, whereas OABSS subsequently worsened in the delayed resolution group. After 6 months after LDR-BT, the delayed resolution group remained significantly higher than the OABSS resolution group with respect to OABSS.

**Figure 3 cancers-15-04048-f003:**
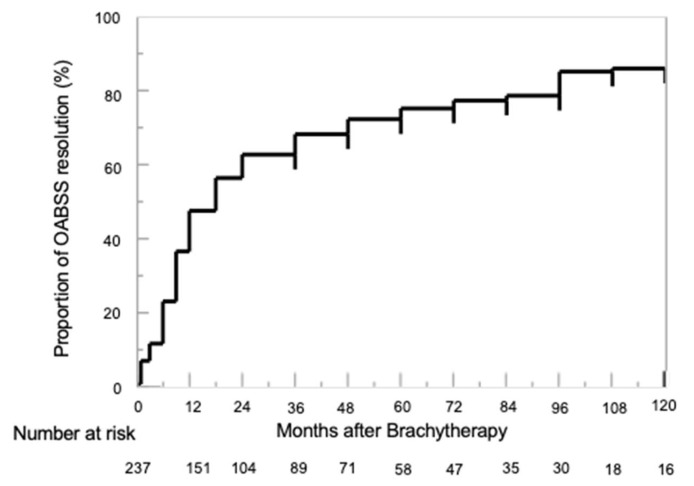
Cumulative resolution of overactive bladder symptom scores was 47.3%, 67.9%, and 74.9% at 1, 3, and 5 years after low-dose-rate brachytherapy with iodin-125.

**Figure 4 cancers-15-04048-f004:**
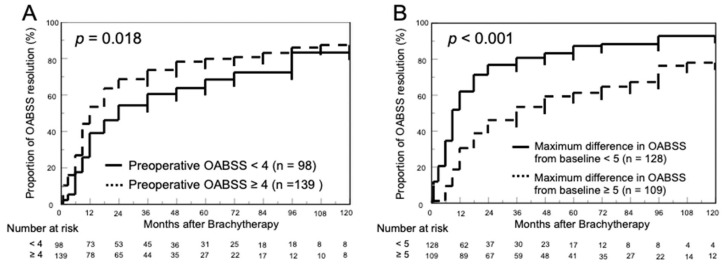
Cumulative resolution rate of overactive bladder symptom score (OABSS) was evaluated. (**A**) The OABSS resolution rates at 1, 3, and 5 years after low-dose-rate brachytherapy (LDR-BT) with iodin-125 were 38.8%, 60.2%, and 68.2% for patients with preoperative OABSS <4 and 53.2%, 73.4%, and 79.6% for patients with ≥4 (*p* = 0. 018; (**A**)). (**B**) The OABSS resolution rates at 1, 3, and 5 years after LDR-BT were 61.7%, 80.5%, and 87.0%, respectively, for patients with a preoperative to maximal change in OABSS of <5 and 30.3%, 53.2%, and 61.1%, respectively, for those with a change of ≥5 (*p* < 0.001; (**B**)).

**Table 1 cancers-15-04048-t001:** Patient characteristics.

Covariates	OABSS Resolution Group (*n* = 192)	Delayed Resolution Group (*n* = 45)	*p*-Value *
Age (year, median, IQR)	66.5 (63.0–71.0)	64.0 (60.0–69.0)	0.023
BMI (median, IQR)	23.6 (21.9–25.3)	23.0 (21.8–25.6)	0.921
Initial PSA (ng/mL, median, IQR)	6.3 (5.0–8.7)	5.4 (4.5–7.6)	0.029
Prostate volume (mL, median, IQR)	24.9 (18.6–34.9)	24.0 (18.5–35.1)	0.664
Biopsy Gleason grade group (number, %)			0.355
1	88 (42.7)	25 (25.9)	
2	65 (34.8)	13 (29.6)	
3	23 (13.4)	3 (22.2)	
4	11 (6.4)	1 (7.4)	
5	5 (2.8)	3 (14.8)	
Clinical T stage (number, %)			0.167
1	134 (69.8)	33 (73.3)	
2	58 (30.2)	11 (24.4)	
3	0 (0)	1 (2.2)	
NCCN risk classification (number, %)			0.555
Low	80 (41.7)	24 (53.3)	
Favorable intermediate	69 (35.9)	14 (31.1)	
Unfavorable intermediate	26 (13.5)	3 (6.7)	
High	16 (8.3)	4 (8.9)	
Very High	1 (0.5)	0 (0)	
Inserted needles (number, median, IQR)	21 (19–24)	22 (20–24)	0.269
Inserted seeds (number, median, IQR)	65 (54–84)	70 (60–85)	0.323
BED (Gy, median, IQR)	192 (173–205)	192 (176–212)	0.590
D90 (Gy, median, IQR)	160 (130–175)	168 (137–183)	0.055
V100 (%, median, IQR)	96 (94–98)	97 (95–98)	0.266
UD30 (Gy, median, IQR)	200 (169–221)	207 (181–239)	0.124
RV100 (cc, median, IQR)	0.25 (0.08–0.79)	0.54 (0.15–0.88)	0.072
Combined ADT (number, %)	160 (83.3)	35 (77.8)	0.389
Supplementary EBRT (number, %)	62 (32.3)	11 (24.4)	0.371
Preoperative maximal flow rate (mL/s, median, IQR)	18 (14–23)	18 (15–22)	0.754
Preoperative PVR (mL, median, IQR)	15 (7–25)	13 (4–20)	0.224
Preoperative IPSS (median, IQR)	5 (3–9)	6 (3–11)	0.693
Preoperative QOL (median, IQR)	2 (1–3)	2 (1–3)	0.826
Preoperative OABSS (median, IQR)	3 (2–5)	2 (1–4)	0.004
Change from baseline to maximum in OABSS (median, IQR)	4 (2–6)	6 (4–8)	<0.001
Time to maximal OABSS (month, median, IQR)	3 (1–3)	3 (1–8)	0.081
Time to the resolution of OABSS (month, median, IQR)	12 (6–24)	-	-
Biochemical recurrence (number, %)	3 (1.6)	1 (2.2)	0.572
Follow-up period (month, median, IQR)	87 (68–120)	87 (72–108)	0.461

IQR, Interquartile range; BMI, body mass index; PSA, prostate-specific antigen; NCCN, National Comprehensive Cancer Network; BED, biologically effective dose; D90, minimum dose received by 90% of the target volume; V100, percentage of target volume receiving a minimum of 100% of the prescribed dose; UD30, minimal dose received by 30% of the urethra; RV100, rectal volume receiving 100% of the prescribed dose; ADT, androgen deprivation therapy; EBRT, external beam radiotherapy; PVR, post-voided residual urine; IPSS, International Prostate Symptom Score; QOL, quality of life due to urinary symptoms; OABSS, overactive bladder symptom score. * Wilcoxon rank-sum test and Fisher’s exact test.

**Table 2 cancers-15-04048-t002:** Multivariable Cox proportional hazard regression analyses for overactive bladder symptom score (OABSS) resolution.

Variable	Univariate		Multivariate	
	HR (95% CI)	*p*-Value	HR (95% CI)	*p*-Value
Age (continuous)	1.023 (0.997–1.050)	0.080	1.014 (0.989–1.041)	0.276
BMI (continuous)	0.994 (0.936–1.054)	0.840		
Initial PSA (continuous)	0.987 (0.954–1.016)	0.393		
Prostate volume (continuous)	1.000 (0.999–1.001)	0.366	1.001 (1.000–1.002)	0.077
BED (continuous)	0.995 (0.989–1.001)	0.109		
UD30 (continuous)	0.998 (0.994–1.001)	0.232	1.000 (0.997–1.004)	0.835
Combined ADT (vs. none)	1.304 (0.904–1.941)	0.160		
Supplementary EBRT (vs. none)	0.971 (0.712–1.309)	0.847		
Preoperative IPSS (continuous)	1.001 (0.972–1.030)	0.941		
Preoperative OABSS (continuous)	1.192 (1.115–1.272)	<0.001	1.155 (1.079–1.234)	<0.001
Change from baseline to maximum in OABSS (continuous)	0.772 (0.721–0.825)	<0.001	0.785 (0.733–0.839)	<0.001

BMI, Body mass index; PSA, prostate-specific antigen; BED, biologically effective dose; UD30, minimal dose received by 30% of the urethra; ADT, androgen deprivation therapy; EBRT, external beam radiotherapy; IPSS, International Prostate Symptom Score; OABSS, overactive bladder symptom score.

## Data Availability

Data and materials are provided in this paper.

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
