# Peer review of "Prognostic Factors for Resolution Delay of Lower Urinary Tract Symptoms in Patients with Prostate Cancer after Low-Dose-Rate Brachytherapy"

_cancers, 2023, doi:10.3390/cancers15164048_

Round 1

Reviewer 1 Report

You have submitted quite a fascinating manuscript and I sure would like to see it printed soon. However, I would suggest extensive revision in combination with re-review for this manuscript.

1) In the discussion, a number of previous reports are cited, but the comparison with the results obtained in this study and their significance are not fully discussed. For example,  a large number of previous studies examining chronological changes after LDR-BT have been reported, but What is the novelty of the results obtained in the present study of yours? I believe readers will have more benefit if you consider the results and discuss them in more detail, such as in how they can be given clinical feedback.

2) You conclude: this study identified independent factors influencing OABSS resolution; low OABSS before LDR-BT and a large change in OABSS from baseline were independent pedictors of delayed OABSS resolution. Certainly, if pre-treatment OABSS is low, alternative treatment options can be considered. However, if a significant change in OABSS after LDR-BT from baseline occurs, how can you feed back to the clinic, even if you can foresee a delayed resolution of OABSS?

3)In Figure 4A, why were they divided into two groups with a preoperative OABSS of less than 4 and more than 4? Why did you divide the maximum change in OABSS into two groups in Figure 4B: less than 5 and greater than 5? Why did you chose the numbers 4 and 5, respectively?

4) Why does OABSS worsen in the Resolution delayed group 6 years after treatment? Please deepen the discussion with your consideration.

Author Response

August 3, 2023

Dear Editor-in-Chief

The Cancers

Thank you very much for the review of our manuscript titled “Prognostic Factors for Resolution Delay of Lower Urinary Tract Symptoms in Patients with Prostate Cancer After Low-dose-rate Brachytherapy.”

We sincerely appreciate all valuable comments and suggestions, which helped us to improve the quality of our manuscript. Our responses to the Reviewers’ comments are described below in a point-to-point manner. Appropriate changes, suggested by the Reviewers, have been introduced to the manuscript (track-changes mode in the red color font). Let me emphasize our full readiness to make any further improvements to the manuscript.

We hope that our manuscript will be acceptable for publication in the Cancers.

We look forward to hearing from you.

Yours sincerely,

Takuya Koie

Department of Urology

Gifu University Graduate School of Medicine

1-1 Yanagido, Gifu, Gifu 501-1194, Japan

TEL.: +81-582-30-6000

Responses to the reviewer's comments

We would like to thank the Reviewers for taking the time and effort necessary to review the manuscript. We sincerely appreciate all the valuable comments and suggestions, which helped us to improve the quality of the manuscript.

Response to Reviewer 1

The authors appreciate the reviewer’s comments. The authors’ point-by-point responses to the comments are given below.

1) In the discussion, a number of previous reports are cited, but the comparison with the results obtained in this study and their significance are not fully discussed. For example, a large number of previous studies examining chronological changes after LDR-BT have been reported, but What is the novelty of the results obtained in the present study of yours? I believe readers will have more benefit if you consider the results and discuss them in more detail, such as in how they can be given clinical feedback.

Response:

The authors have added the following sentence on line 239:

In the current study, PV, dosimetric parameters and preoperative IPSS were not associated with late GU toxicity for storage symptom.

The authors have revised the following sentences on line 260:

In the present study, the OABSS resolution rate was 47.3% at 12 months and 67.9% at 36 months, which was relatively poor compared to the results of previous studies. At our institution, patients with severe LUTS such as Qmax <10 mL/s, high IPSS or high OABSS were excluded from LDR-BT, suggesting that these results may be due to selection bias.

The authors have revised the following sentences on line 304:

In the present study, which allowed for long-term observation, OABSS resolution was not associated with additional EBRT treatment, concomitant ADT, BED, or UD30. Although these factors might temporarily worsen LUTS in the early postoperative period, OABSS tended to improve gradually over time, suggesting that they might not affect OABSS after ≥ 4 years postoperatively.

2) You conclude: this study identified independent factors influencing OABSS resolution; low OABSS before LDR-BT and a large change in OABSS from baseline were independent predictors of delayed OABSS resolution. Certainly, if pre-treatment OABSS is low, alternative treatment options can be considered. However, if a significant change in OABSS after LDR-BT from baseline occurs, how can you feed back to the clinic, even if you can foresee a delayed resolution of OABSS?

Response:

The authors have added the following sentence on line 283:

If the OABSS changed significantly from baseline after LDR-BT, it seemed necessary to inform the patient that conservative symptoms might continue for some time.

3) In Figure 4A, why were they divided into two groups with a preoperative OABSS of less than 4 and more than 4? Why did you divide the maximum change in OABSS into two groups in Figure 4B: less than 5 and greater than 5? Why did you choose the numbers 4 and 5, respectively?

Response:

The authors already described the following sentence on line 195:

Patients were divided into two groups based on the median values of preoperative OABSS and the difference between the maximum OABSS and baseline.

4) Why does OABSS worsen in the Resolution delayed group 6 years after treatment? Please deepen the discussion with your consideration.

Response:

The authors have added the following sentences on line 272:

In this study, OABSS worsened only in the delayed resolution group 6 years after treatment, and the reason for this was unclear; although changes in bladder function after LDR-BT could be one cause, the results of this study suggest that age is not associated with worsening OABSS. Therefore, it is possible that radiation therapy caused some damage to the normal tissues of the bladder. However, it is difficult to determine when and to what extent bladder tissue was affected in patients who received LDR-BT. In fact, there were no reports that examined in detail the effects on normal bladder tissue after radiotherapy. Further studies may be necessary in the future.

Reviewer 2 Report

interesting manuscript on LDR-BT influence on OABSS and prognostic factors for delayed resolution of storage symptoms

Introduction - nicely written paragraph, clearly presenting the background and the available literature on this topic

row 53 - this sentence is unclear? re-check? "curative therapy for low-risk PCa and EBRT alone"

Material and methods - concisely written paragraph - no remarks

Results - the main advantage of the study is the long period of follow-up - 88.7 months. 

Paragraph 3.2 and 3.3 - very good visual presentation of the results, 

another strength of the study is the usage of multivariate analysis which identifies lower baseline OABSS and significant change in OABSS post LDR-BT as significant predictor for delayed resolution of storage symptoms

Discussion - elegantly implements and compare authors experience with published literature 

- in this reviewer`s opinion there is heterogeneity of radiotherapy type in this cohort - strands vs loose seeds from different manufacturers, LDR as monotherapy vs combination with EBRT - this leads to a potential bias and should be avoided in future prospective studies on the topic, as well as the other limitations mentioned by the authors

Conclusions - results support the usage of low baseline OABSS and significant increase of OABSS post LDR as clinical predictors to delayed resolution of storage symptoms

The conclusions are adequately substantiated and represent a solid base to further investigation on the subject. The references are appropriate and relevant to the subject and my recommendation is to accept this manuscript for publication after discussion on the above-mentioned topics.

Author Response

August 3, 2023

Dear Editor-in-Chief

The Cancers

Thank you very much for the review of our manuscript titled “Prognostic Factors for Resolution Delay of Lower Urinary Tract Symptoms in Patients with Prostate Cancer After Low-dose-rate Brachytherapy.”

We sincerely appreciate all valuable comments and suggestions, which helped us to improve the quality of our manuscript. Our responses to the Reviewers’ comments are described below in a point-to-point manner. Appropriate changes, suggested by the Reviewers, have been introduced to the manuscript (track-changes mode in the red color font). Let me emphasize our full readiness to make any further improvements to the manuscript.

We hope that our manuscript will be acceptable for publication in the Cancers.

We look forward to hearing from you.

Yours sincerely,

Takuya Koie

Department of Urology

Gifu University Graduate School of Medicine

1-1 Yanagido, Gifu, Gifu 501-1194, Japan

TEL.: +81-582-30-6000

Responses to the reviewer's comments

We would like to thank the Reviewers for taking the time and effort necessary to review the manuscript. We sincerely appreciate all the valuable comments and suggestions, which helped us to improve the quality of the manuscript.

Response to Reviewer 2

The authors appreciate the reviewer’s comments. The authors’ point-by-point responses to the comments are given below.

1) Introduction - nicely written paragraph, clearly presenting the background and the available literature on this topic

row 53 - this sentence is unclear? re-check? "curative therapy for low-risk PCa and EBRT alone"

Response:

Thank you for your kind comments.

The authors have deleted the following part on line 54:

Among them, low-dose-rate brachytherapy (LDR-BT) with iodine-125 has been considered a curative therapy for low-risk PCa as well as EBRT alone [4].

2) Material and methods - concisely written paragraph - no remarks

Response:

Thank you for your kind comments.

3) Results - the main advantage of the study is the long period of follow-up - 88.7 months.

Paragraph 3.2 and 3.3 - very good visual presentation of the results,

another strength of the study is the usage of multivariate analysis which identifies lower baseline OABSS and significant change in OABSS post LDR-BT as significant predictor for delayed resolution of storage symptoms

Response:

Thank you for your kind comments.

4) Discussion - elegantly implements and compare authors experience with published literature

- in this reviewer`s opinion there is heterogeneity of radiotherapy type in this cohort - strands vs loose seeds from different manufacturers, LDR as monotherapy vs combination with EBRT - this leads to a potential bias and should be avoided in future prospective studies on the topic, as well as the other limitations mentioned by the authors

Response:

The authors have added the following sentence on line 317:

Forth, there is heterogeneity of radiotherapy type in this cohort, namely linked seeds or loose seeds, LDR-BT monotherapy or combination therapy with EBRT. This leads to a potential bias and should be avoided in future prospective studies.

5) Conclusions - results support the usage of low baseline OABSS and significant increase of OABSS post LDR as clinical predictors to delayed resolution of storage symptoms

Response:

Thank you for your kind comments.

Round 2

Reviewer 1 Report

The discussion part has been improved and the content is beneficial for the readers.

Just one minor revision. In the legends of Figure 4, you show the results at 1, 3, and 5 years, but the horizontal scale of Figure 4 is every 20 months, which is confusing. It would be better to correct this point.

Author Response

Aug 9, 2023

Dear Editor-in-Chief

The Cancers

Dear Editor

Thank you very much for the review of our manuscript titled “Prognostic Factors for Resolution Delay of Lower Urinary Tract Symptoms in Patients with Prostate Cancer After Low-dose-rate Brachytherapy.”

We sincerely appreciate all valuable comments and suggestions, which helped us to improve the quality of our manuscript. Our responses to the Reviewers’ comments are described below in a point-to-point manner. Appropriate changes, suggested by the Reviewers, have been introduced to the manuscript. Let me emphasize our full readiness to make any further improvements to the manuscript.

We hope that our manuscript will be acceptable for publication in the Cancers.

We look forward to hearing from you.

Yours sincerely,

Takuya Koie

Department of Urology

Gifu University Graduate School of Medicine

1-1 Yanagido, Gifu, Gifu 501-1194, Japan

TEL.: +81-582-30-6000

Responses to the reviewer's comments

We would like to thank the Reviewers for taking the time and effort necessary to review the manuscript. We sincerely appreciate all the valuable comments and suggestions, which helped us to improve the quality of the manuscript.

Response to Reviewer 2

The authors appreciate the reviewer’s comments. The authors’ point-by-point responses to the comments are given below.

1) The discussion part has been improved and the content is beneficial for the readers. Just one minor revision. In the legends of Figure 4, you show the results at 1, 3, and 5 years, but the horizontal scale of Figure 4 is every 20 months, which is confusing. It would be better to correct this point.

Response:

The authors have revised the Figure 3 and 4 according to the reviewer’s recommendation.